Original research

# Cost-effectiveness of a dietary and physical activity intervention in adolescents: a prototype modelling study based on the Engaging Adolescents in Changing Behaviour (EACH-B) programme

Neelam Kalita [1], Keith Cooper [1], Janis Baird,[2,3] Kathryn Woods-Townsend,[3,4] Keith Godfrey,[2,3] Cyrus Cooper,[2,5] Hazel Inskip [2,3] Mary Barker,[2,3,6] Joanne Lord,[1] the EACH-B study group

For numbered affiliations see end of article.

**Correspondence to**
Neelam Kalita;
N.Kalita@soton.ac.uk

## ABSTRACT

**Objective** To assess costs, health outcomes and cost-effectiveness of interventions that aim to improve quality of diet and level of physical activity in adolescents.

**Design** A Markov model was developed to assess four potential benefits of healthy behaviour for adolescents: better mental health (episodes of depression and generalised anxiety disorder), higher earnings and reduced incidence of type 2 diabetes and adverse pregnancy outcomes (in terms of preterm delivery). The model parameters were informed by published literature. The analysis took a societal perspective over a 20-year period. One-way and probabilistic sensitivity analyses for 10 000 simulations were conducted.

**Participants** A hypothetical cohort of 100 adolescents with a mean age of 13 years.

**Interventions** An exemplar school-based, multicomponent intervention that was developed by the Engaging Adolescents for Changing Behaviour programme, compared with usual schooling.

**Outcome measure** Incremental cost-effectiveness ratio (ICER) as measured by cost per quality-adjusted life-year (QALY) gained.

**Results** The exemplar dietary and physical activity intervention was associated with an incremental cost of £123 per adolescent and better health outcomes with a mean QALY gain of 0.0085 compared with usual schooling, resulting in an ICER of £14 367 per QALY. The key model drivers are the intervention effect on levels of physical activity, quality-of-life gain for high levels of physical activity, the duration of the intervention effects and the period over which effects wane.

**Conclusions** The results suggested that such an intervention has the potential to offer a cost-effective use of healthcare-resources for adolescents in the UK at a willingness-to-pay threshold of £20 000 per QALY. The model focused on short-term to medium-term benefits of healthy eating and physical activity exploiting the strong evidence base that exists for this age group. Other benefits in later life, such as reduced cardiovascular risk, are

## STRENGTHS AND LIMITATIONS OF THIS STUDY

⇒ The model was developed, based on currently available evidence, to assess the costs and health benefits of an exemplar school-based, multicomponent intervention to improve adolescents' diet and increase their levels of physical activity.
⇒ The economic evaluation followed published best-practice guidance.
⇒ In the absence of effectiveness results from the ongoing Engaging Adolescents in CHanging Behaviour trial, the treatment effect of the intervention is obtained from a published systematic review.
⇒ The model does not include long-term health outcomes, such as for cardiovascular diseases or different types of cancer.
⇒ There remains uncertainty around key model assumptions related to the duration of benefits of the intervention.

more sensitive to assumptions about the persistence of behavioural change and discounting.

**Trail registration number** ISRCTN74109264.

## INTRODUCTION

Poor diet and lack of physical activity increase the risk of non-communicable diseases (NCDs), including cardiovascular diseases, type 2 diabetes and some cancers such as breast, colon and endometrial, in part by contributing to overweight and obesity.[1 2] Adolescence, the life stage between childhood and adulthood, is a critical period for the development of health and disease in later life.[3 4] Compared with other age groups, adolescents have the unhealthiest diets and most (over 80%) fail to meet

national guidelines for physical activity.[5–7] Furthermore, the proportion meeting recommended levels of physical activity has been declining, particularly among girls (from 14% to 8%, 2008–2012).[8]

The disease burden of poor diet and physical inactivity on healthcare services is significant. In the UK, poor diet and physical inactivity cost £7 billion to the National Health Service (NHS) annually.[9] Meeting current dietary recommendations would reduce years-of-life lost to coronary heart disease by 2 million, stroke by 400 000 and type 2 diabetes by 19 000 over 20 years.[10]

Health behaviours in adolescence track into adulthood.[5 9 11 12] Therefore, suboptimal diet and body composition in adolescence not only affect immediate physical and mental health but also increase the risk of NCDs in later life. Developmental plasticity in adolescence means that interventions to improve diet and levels of physical activity have the potential to reduce the trajectory of NCD risk over the life course.[13] While many adolescents find it difficult to engage with the long-term consequences of health behaviour, motivated and engaged adolescents can improve their health behaviours.[14 15] Evidence suggests that school-based interventions that offer combinations of peer-modelling, social support and choice, may be effective in improving diet and physical activity among adolescents.[16–18] Furthermore, there is an increasing use of digital platforms by adolescents. According to 2018 estimates, 83% of 12–15 years olds in the UK own smartphones, with 99% spending an average of 20 hours per week online.[19 20] With an explosion in the use of such platforms to influence health behaviours in young people, they have potential as a complementary feature in complex interventions that aim to influence health behaviour in adolescents.[20]

Within this framework, a research programme Engaging Adolescents in CHanging Behaviour (EACH-B) was designed to develop and test an intervention to encourage UK-based school students, aged 12–13 years, to adopt healthy behaviours such as eating better and exercising more (Trial registration: ISRCTN74109264). EACH-B involves a cluster randomised controlled trial as a test of intervention effectiveness. Further details of the trial design are given elsewhere.[20] The 'LifeLab Plus' intervention developed as part of this programme is a complex three-part programme that comprises: (1) an education module that teaches school students the science behind health messages through a 2-week module with a 'hands-on' practical 1-day visit to a teaching laboratory at University Hospital Southampton or in school while COVID-19 restrictions apply; (2) training for teachers in skills to support behaviour change and (3) access to a specially designed, interactive smartphone app with game features.

There is an emerging interest in identifying and developing interventions for improving diet and physical activity levels in adolescents. Taking this into consideration, we developed an illustrative decision-analytical model to assess the health benefits, costs and cost-effectiveness of a multicomponent intervention such as LifeLab Plus. This prototype model is designed to investigate how changes in diet quality and physical activity could affect future health outcomes and costs. We used cost data from the EACH-B programme and effectiveness estimates for similar interventions from published literature. The aims of this paper are to describe the structure, assumptions and parameters for our prototype model, to present preliminary (prior) estimates of cost-effectiveness based on currently available information and explore the sensitivity of results to key uncertainties. The model will be updated when results from the EACH-B trial are available.

## METHODS

We developed a prototype de novo Markov model to estimate the costs, benefits and cost-effectiveness of school-based interventions that aim to improve diet quality and levels of physical activity compared with usual schooling for a cohort of adolescents. The model focused on four potential short to medium-term benefits of healthy eating and physical activity: better mental health (episodes of depression and generalised anxiety disorder), higher earnings and reduced incidence of type 2 diabetes and adverse pregnancy outcomes (in terms of preterm delivery). The model assumed that improved diet quality and increased physical activity would impact these four aspects of health via a reduction in body mass index (BMI). Discussion with key project stakeholders reiterated these four benefits as the most relevant obesity-related effects in this population.

The model did not include later life impacts on cardiovascular disease or other chronic diseases as the impact on these outcomes of an intervention undertaken as an adolescent is uncertain. The model also investigated independent effects of physical activity on diabetes and depression (ie, direct impacts not mediated by BMI). Information relating to epidemiology, mortality, effectiveness, health-related quality of life (HRQoL) and costs was obtained from a variety of sources to inform the model parameters and assumptions.

### Structuring the model
#### Population
A cohort of 100 adolescents with an equal proportion of boys and girls and a mean age of 13 years entered the model. The exemplar intervention is based on LifeLab Plus[20] and compared with usual schooling.

#### Model states
The Markov model consisted of three health states as described below (figure 1). Outcomes associated with mental health, loss of earnings and adverse pregnancy outcome were incorporated as model events.
► Healthy: Adolescents enter the model in this state where they are assumed to have no other comorbidities including type 2 diabetes or mental health problems. In each model cycle, a proportion will develop

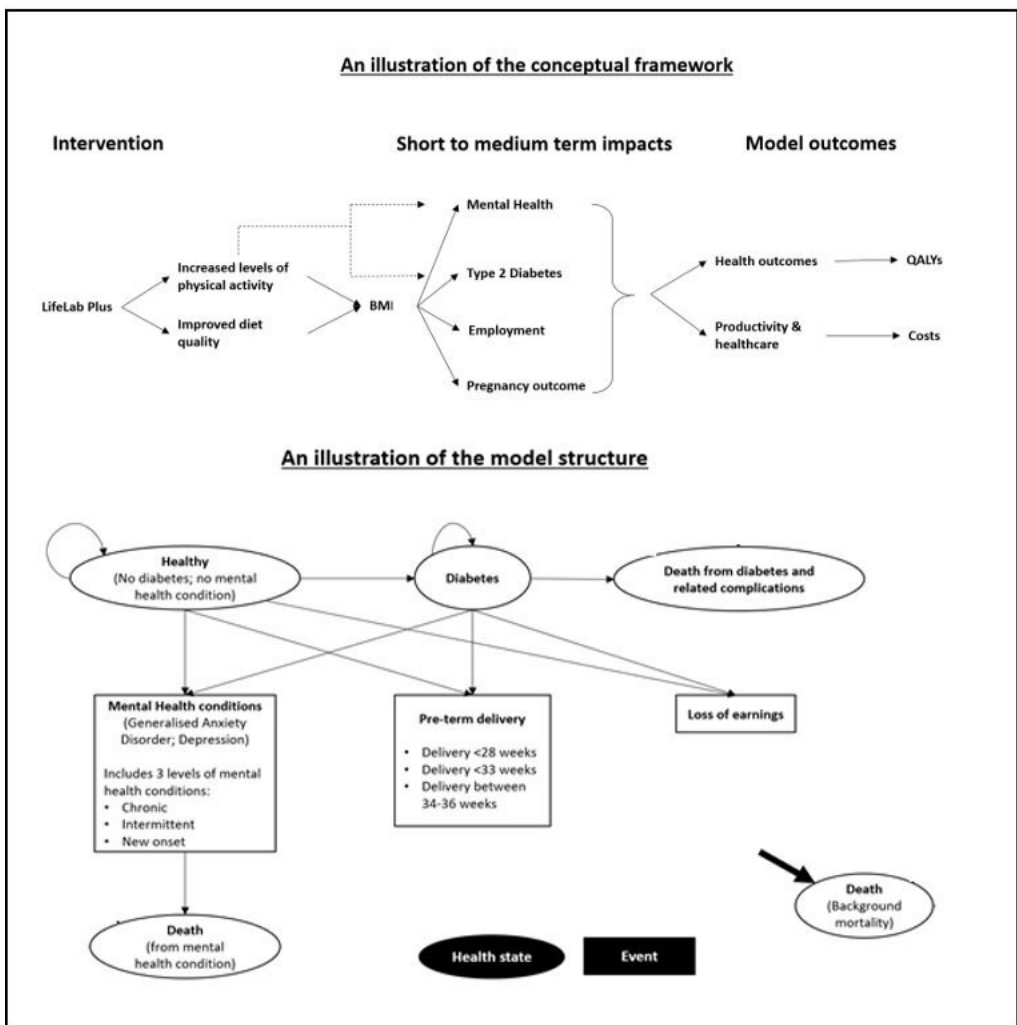

**Figure 1** Illustrations of the conceptual framework for the cost-effectiveness model and the model structure showing transition of the modelled population through the modelled health states. QALY, quality-adjusted life-year.

type 2 diabetes and move to the 'Diabetes' state or they may experience—mental health events, pre-term births (girls only) or loss of earnings. The latter three are treated as transient states.

► Diabetes: A proportion of adolescents with type 2 diabetes may enter the transient states, as stated above.

► Death: This is an absorbing state. Adolescents may transition to this state either through general population mortality or death from—mental health conditions or—type 2 diabetes and related complications.

Mental health encompasses a wide spectrum of conditions. Therefore, a pragmatic approach was adopted to include the two most prevalent mental health conditions for adolescents: clinical depression (henceforth, referred as depression) and General Anxiety Disorder (GAD). These conditions were categorised as: chronic (history of persistent mental illness), intermittent (experiencing intermittent episodes) and new onset (a one-time episode). Adverse pregnancy outcome was defined by preterm delivery categorised as: extremely preterm (delivery <28 weeks); very preterm (delivery <33 weeks) and moderately preterm (delivery between 34 and 36

weeks). The pregnancy outcome was applicable for girls only.

Each health state and event are associated with an impact on HRQoL and excess cost. The model included intervention cost associated with LifeLab Plus and healthcare costs associated with type 2 diabetes, mental health events, preterm delivery and loss of earnings due to obesity. In each model cycle, the total costs and quality-adjusted life-years (QALYs) are calculated by multiplying the individual costs and HRQoL by the number of people in the cohort still alive for each of the intervention and control arms. The total lifetime costs and QALYs are calculated by aggregating the costs and QALYs for all cycles.

### Persistence of effects and time horizon

Adolescents were assumed to receive the intervention at the model entry, the duration of which lasted for a year. Although there is good epidemiological evidence of long-term tracking of health behaviour,[5 9 11 12] school-based trials rarely follow-up for more than a year.[1 21 22] The persistence of intervention effect in terms of behaviour

change is therefore uncertain. Hence, we take a conservative approach and focus on potential impacts of improved diet and physical activity from such an intervention likely to manifest in the short term to medium term: up to a maximum time horizon of 20 years. However, due to the uncertainty, the treatment effect on behaviour was not assumed to last for 20 years. Based on our experts' judgement, for the base case the effect was assumed to sustain for 4 years, waning to no effect over a period of 10 years. We explored alternative assumptions about the persistence of effects on behaviour after trial follow-up.

The model time horizon was 20 years with a yearly cycle length. The time horizon was considered appropriate as the mean age of adolescents entering the model was 13 years as in the EACH-B trial. Costs and outcomes were half-cycle corrected.

### Populating the model
Targeted literature searches were conducted to identify sources to inform model parameters. These are discussed below and in online supplemental appendices A and B.

### Epidemiological data
Data on the relationship between mean BMI, age and sex was taken from Health Survey for England.[23] For adolescents aged ≤19 years, BMI z-scores are normally used. Therefore, we rescaled the values to relate relative risks to BMI z-scores, where:

BMI z-score = (observed value − median value of the reference population)/SD value of reference population.

Physical activity levels for children aged 13–15 years were taken from Health Survey England, 2015.[24] The incidence of type 2 diabetes in the UK was based on an analysis of longitudinal electronic health records in the Health Improvement Network primary care database.[25] The prevalence of depressive episodes and GAD was taken from the Adult Psychiatric Morbidity Survey 2014[26] and from Mental Health of Children and Young people in England 2017.[27] The proportion in each category was assumed as follows: 17% had a chronic and 40% had a fluctuating (intermittent) course, while 43% remitted (new one-time episode).[28] Those individuals with depression or anxiety are at higher risk of suicide than the general population.[29] The excess death rate for those with depression and anxiety was calculated by using the suicide rate in the UK from Office of National Statistics (ONS) 2017 and applying a relative risk of 10.9 for depression and anxiety.[29] The proportions of preterm deliveries, obtained from ONS 2017 data, were assumed as follows: 0.5% of total births as extremely preterm (<28 weeks), 1.2% very preterm birth (28 to <33 weeks) and 6.3% moderately preterm birth (33–36 weeks), respectively.[30]

### Relationship between BMI and risks of health events
The economic model assumed a positive correlation between increased BMI and the risks of type 2 diabetes,

depression and GAD, preterm delivery and loss of earnings. We fitted equations to the BMI relative risks. HRs, obtained from the Medicare Current Beneficiary Survey 1991–2010, were used to estimate the increased risk of individuals with higher BMI developing type 2 diabetes.[31] The odds of depression and GAD in obese and overweight adolescents compared with normal-weight adolescents were obtained from Sutaria et al.[32] This systematic review included 22 observational studies published between 2000 and 2017, representing 143 603 children. The relative risk of preterm birth for mothers with overweight and obesity was obtained from McDonald et al.[33] This was assumed to be the same for all three categories of preterm births. The loss of earnings which is applied to change in BMI over time is estimated annually over the model time horizon. It is scaled according to the population age to factor in average income by age.

### Relationship between physical activity and risk of health event
The direct effect of physical activity on developing depression is modelled independent of the effect of physical activity via BMI. The OR of developing depression was assumed to be 0.83 (95% CI 0.79 to 0.88) in those with high levels of physical activity compared with those with lower levels.[34] Furthermore, an increase from being inactive to achieving 150 min of moderate-intensity physical activity per week was assumed to lower the risk of type 2 diabetes incidence by 26%, after adjustment for body weight.[35] The pooled OR between type 2 diabetes and risk of depression was 1.33 (95% CI 1.18 to 1.51).[36]

### Intervention effect
The intervention effect was based on three systematic reviews and meta-analyses that estimated the overall effects of school-based obesity prevention interventions.[37–39] The results of the meta-analyses were found to be significantly different between groups based on BMI (− 0.17 (95% CI− 0.29 to −0.06) kg/m$^2$)[38] and BMI z-score (− 0.06 (95% CI −0.10 to −0.03))[39] for multicomponent interventions including physical activity, health education and dietary improvement. The effect of school-based intervention was assumed to increase the level of moderate or vigorous physical activity in children and adolescents by 4.84 min/day (95% CI −0.94 to 10.61).[40]

### Mortality
General population mortality, adjusted for age and gender, was based on ONS 2020.[41] An HR of mortality of 2.98 was applied for individuals with type 2 diabetes and aged between 35 and <65 years.[42]

### Quality of life
European Quality of Life 5 Dimensions 3 Level Version (EQ-5D-3L) estimates adjusted for age, gender and BMI were used to estimate quality of life (table 1). These estimates were obtained from 14 117 participants aged ≥16 years from the Health Survey for England 2008.[43] Adults with diabetes were assumed to have a disutility of −0.161.[44] We estimated the disutility by comparing the mean

**Table 1**  Input parameters used in the model

| Parameter | Value | SD/SE | 95% CI-low | 95% CI-high | Distribution | Reference |
|---|---|---|---|---|---|---|
| Utility inputs | | | | | | |
| Disutility for diabetes for 13–18 years old | −0.087 | 0.036 | 0.016 | 0.158 | Beta | 45 |
| Disutility for diabetes | −0.161 | 0.040 | 0.240 | 0.080 | Beta | 44 |
| Disutility for preterm babies <33 weeks | −0.066 | 0.016 | 0.098 | 0.035 | Beta | 48 |
| Disutility for parents of preterm babies <33 weeks | −0.094 | 0.019 | 0.057 | 0.131 | Beta | Assumption |
| Disutility for depression | −0.188 | 0.038 | 0.114 | 0.262 | Beta | 46 |
| Disutility for chronic mental health condition | −0.448 | 0.036 | 0.377 | 0.519 | Beta | 47 |
| Disutility for intermittent mental health condition | −0.188 | 0.038 | 0.114 | 0.262 | Beta | 46 |
| Disutility for new cases mental health condition | −0.094 | 0.019 | 0.057 | 0.131 | Beta | Assumption |
| Quality-of-life gain for high activity | 0.020 | 0.002 | 0.016 | 0.024 | Beta | |
| Direct costs | | | | | | |
| Type 2 diabetes | £622 | £62 | £500 | £744 | Gamma | 50 |
| Mental health condition | £1334 | £383 | £585 | £2307 | Gamma | 51 |
| Mental health condition-chronic | £4233 | £629 | £3001 | £7238 | Gamma | 51 |
| Mental health condition-intermittent | £1334 | £383 | £585 | £2307 | Gamma | 51 |
| Mental health condition-new onset | £667 | £191 | £292 | £1042 | Gamma | 51 |
| Pre-term birth <28 weeks | £25 452 | £2169 | £21 201 | £29 704 | Gamma | 52 53 |
| Preterm birth 28–33 weeks | £13 073 | £1114 | £10 889 | £15 256 | Gamma | 52 53 |
| Preterm birth, 34–36 weeks | £4157 | £1020 | £2157 | £6157 | Gamma | 52 53 |
| Indirect costs | | | | | | |
| Type 2 diabetes | £4116 | £412 | £3309 | £4923 | Gamma | 50 |
| Mental health condition | £223 | £64 | £98 | £348 | Gamma | 51 |
| Mental health condition-chronic | £707 | £203 | £310 | £1105 | Gamma | 51 |
| Mental health condition-intermittent | £223 | £64 | £98 | £348 | Gamma | 51 |
| Mental health condition-new onset | £112 | £32 | £49 | £174 | Gamma | 51 |
| Pre-term birth <28 weeks | £397 | £114 | £174 | £619 | Gamma | 52 53 |
| Preterm birth 28–33 weeks | £119 | £34 | £52 | £185 | Gamma | 52 53 |
| Preterm birth, 34–36 weeks | £103 | £29 | £45 | £161 | Gamma | 52 53 |
| Cost of the intervention | £155 | £15 | £124 | £185 | Gamma | Estimated |

general population EQ-5D score with that for people with diabetes. A disutility of −0.087 is used for adolescents with type 2 diabetes based on a Swedish cohort of adolescents aged 13–18 years.[45] A utility decrement of 0.188 was assumed for those with intermittent episodes of mental health conditions[46]; a decrement of 0.488 for those with persistent/chronic depression[47]; a decrement of 0.094 (half of the decrement for intermittent depression) was assumed for those with a new episode of depression. For preterm delivery, a mean utility decrement of 0.066 was applied throughout the model time horizon. This was based on a systematic review and meta-analysis for health utility values associated with preterm birth where all but one study used Health Utilities Index (HUI) Mark 2 (HUI2) or Mark 3 (HUI3) measures as their primary health utility assessment method.[48] We found no evidence for quality-of-life loss in parents of preterm babies. Therefore, we assumed that the quality-of-life decrement would be like intermittent mental health condition and lasts for the first 2 years.

## Costs
We describe the costs below and summarise in table 1.

### Intervention costs
The intervention cost for our illustrative analysis was based on LifeLab Plus. Further information on resources used for delivering LifeLab are in online supplemental appendix B. The cost of the app delivered as part of the intervention, was incorporated as a capital cost and was assumed to last 10 years and be used in 10 centres. Similarly, the cost of setting up LifeLab Plus in a different centre was assumed to consist of 1 year's staff costs and last for 10 years. Maintenance costs were estimated at 25% of the development cost per year. Overheads were included according to the rates used in Curtis et al[49]: direct overheads based on 29% of direct care salary costs and indirect overheads based on 16% direct care salary costs.

### Health state costs
Both direct and indirect costs relating to individuals with type 2 diabetes,[50] depression,[51] preterm birth[52] and loss of

earnings were included. Indirect costs included the effect of depression on income and productivity. Costs were updated to 2019 prices using the hospital and community health services index.[49]

Direct health costs and indirect societal and productivity costs were estimated using a top-down approach. We assumed that individuals with type 2 diabetes would not incur costs for complications as these are likely to affect individuals who have diabetes for a longer period. For costs associated with preterm delivery, total societal costs for children born at 32–33 weeks and 34–36 weeks gestation from birth to 24 months were based on a study that compared these costs to those for children born at full term.[53] The difference in costs for children born <28 weeks and 28–33 weeks were assumed to be same when compared with those born at full term.

The lifetime indirect costs for overweight and obesity in childhood and adolescence are based on a study by Hamilton et al.[54] Mean total lifetime healthcare and productivity costs were estimated in Irish Euros, which were converted to GBP (£) and adjusted according to the average wage in the UK.

### Validation
The structure of the prototype model was validated by the study team comprising epidemiologists, statisticians, trialists, public health experts and health economists. Internal validity of the model was established ensuring that the model predictions were consistent with the model specification.

### Cost-effectiveness analysis
Costs and effects were estimated in accordance with the National Institute for Health and Care Excellence (NICE) Reference case,[55] except for the perspective adopted; we used a societal perspective that included both direct and indirect costs. Costs and health effects (QALYs) were discounted at 3.5% per year. The incremental cost-effectiveness ratio (ICER) was estimated as a ratio of the incremental costs of the intervention (LifeLab Plus) relative to the comparator (usual schooling) to the incremental QALYs of the intervention relative to the comparator. The intervention was considered cost-effective if the ICER was below the threshold of £20 000 per QALY gained, the lower threshold usually considered for the English NHS by NICE.[55]

### Sensitivity analyses
To assess the uncertainty around the model predictions, we conducted deterministic sensitivity analysis (DSA), probabilistic sensitivity analyses (PSA) and scenario analyses. For the DSA, input parameters were varied by 95% CIs (where available) or varied by 10% of the mean. Monte Carlo simulations of 10 000 iterations were run for the PSA to assess the combined effects of input parameter uncertainties where parameters were simultaneously sampled within a specified distribution. We used standard distributions for the PSA, as recommended by Briggs et al.[56]: effectiveness parameters were assigned beta and lognormal distributions; utilities were assumed to follow a beta distribution; and costs were assigned a gamma distribution (table 1 and further details in online supplemental appendix A). Scenario analyses were

conducted to assess structural uncertainties related to model assumptions. Uncertainty about the sustainability of the intervention effect was assessed by varying the duration of intervention effect and its waning period.

The model was developed and implemented in Microsoft Excel.

### Patient and public involvement
The research questions addressed in the overarching research programme EACH-B were informed by public involvement. Furthermore, representatives from public involved in the EACH-B research programme were presented the conceptual framework, modelling approaches and invited to comment.

## RESULTS
### Base-case analysis
For the base case, the exemplar intervention based on the multicomponent LifeLab Plus was associated with higher costs and better health outcomes (more QALYs) compared with usual schooling. LifeLab Plus was associated with a mean QALY gain of 0.0085 at an incremental cost of £123 per person compared with usual schooling, resulting in an ICER of £14 367 per QALY (table 2).

### Sensitivity analyses
The results of the one-way sensitivity analyses are presented in the Tornado plot (figure 2). Parameters including the intervention effect on levels of physical activity (expressed in terms of minutes of moderate or vigorous physical activity), quality-of-life gain for high levels of physical activity, duration of the intervention effect and duration of the treatment waning period had the highest impact on the cost-effectiveness results. Other parameters such as effect of physical activity on BMI, time horizon and intervention costs also influenced the base case results, but to a lesser extent. The results of the PSA are presented as a cost-effectiveness scatter plot in figure 3. All the iterations lie in the North-East quadrant of the cost-effectiveness plane, thereby indicating that the intervention is likely to produce health benefits at an additional cost. At a willingness-to-pay threshold of £20 000 per QALY gained, the probability of LifeLab Plus being cost-effective was 69% compared with usual schooling (online supplemental appendix C).

## DISCUSSION
### Main findings
Based on our prototype economic model, we estimate that a multicomponent intervention to improve dietary quality and physical activity, such as LifeLab Plus, is likely to be considered cost-effective under conventional willingness-to-pay thresholds of £20 000–£30 000 per QALY gained in the UK. For our base case, the duration of the treatment effect was assumed to sustain for 4 years with the effect waning over a further period of 10 years. Our sensitivity analyses showed that if the duration of the treatment effect was not sustained to this extent, the intervention

**Table 2** Incremental cost-effectiveness ratios in the base case and scenario analyses

| | Total cost (£) | Total QALYs | Incremental cost (£) | Incremental QALY | ICER |
|---|---|---|---|---|---|
| Base case analysis | | | | | |
| Usual schooling | £4416 | 13.07 | | | |
| LifeLab Plus | £4539 | 13.08 | £123 | 0.0085 | £14 367 |
| Perspective: NHS | | | | | |
| Usual schooling | £2001 | 13.07 | | | |
| LifeLab Plus | £2141 | 13.08 | £140 | 0.0085 | £16 339 |
| Costs discounted at 3.5%; QALYs at 1.5% per annum | | | | | |
| Usual schooling | £4416 | 15.76 | | | |
| LifeLab Plus | £4539 | 15.77 | £123 | 0.0097 | £12 640 |
| Costs and QALYs discounted at 1.5% | | | | | |
| Usual schooling | £5813 | 15.76 | | | |
| LifeLab Plus | £5930 | 15.77 | £117 | 0.0097 | £12 044 |
| Time horizon: 5 years | | | | | |
| Usual schooling | £488 | 4.25 | | | |
| LifeLab Plus | £633 | 4.25 | £145 | 0.0043 | £33 423 |
| Time Horizon: 10 years | | | | | |
| Usual schooling | £1132 | 7.76 | | | |
| LifeLab Plus | £1265 | 7.77 | £134 | 0.0071 | £18 851 |
| Time Horizon: 30 years | | | | | |
| Usual schooling | £11 137 | 16.67 | | | |
| LifeLab Plus | £11 258 | 16.67 | £121 | 0.0088 | £13 719 |
| Time Horizon: 40 years | | | | | |
| Usual schooling | £18 417 | 19.05 | | | |
| LifeLab Plus | £18 537 | 19.06 | £120 | 0.0089 | £13 455 |
| Duration of treatment effect: 1 year and treatment waning: 2 years | | | | | |
| Usual schooling | £4416 | 13.07 | | | |
| LifeLab Plus | £4566 | 13.08 | £150 | 0.0030 | £49 758 |
| Duration of treatment effect: 1 years and treatment waning: 5 years | | | | | |
| Usual schooling | £4416 | 13.07 | | | |
| LifeLab Plus | £4563 | 13.08 | £147 | 0.0043 | £34 089 |
| Duration of treatment effect: 2 years and treatment waning: 5 years | | | | | |
| Usual schooling | £4416 | 13.07 | | | |
| LifeLab Plus | £4560 | 13.08 | £144 | 0.0051 | £28 034 |
| Duration of treatment effect: 5 years and treatment waning: 5 years | | | | | |
| Usual schooling | £4416 | 13.07 | | | |
| LifeLab Plus | £4553 | 13.08 | £137 | 0.0075 | £18 218 |
| Duration of treatment effect: 5 years and treatment waning: 10 years | | | | | |
| Usual schooling | £4416 | 13.07 | | | |
| LifeLab Plus | £4529 | 13.08 | £113 | 0.0092 | £12 184 |
| Duration of treatment effect: 10 years and treatment waning: 10 years | | | | | |
| Usual schooling | £4416 | 13.07 | | | |
| LifeLab Plus | £4432 | 13.09 | £16 | 0.0125 | £1257 |
| Duration of treatment effect: 15 years and treatment waning: 5 years | | | | | |
| Usual schooling | £4416 | 13.07 | | | |
| LifeLab Plus | £4308 | 13.09 | -£108 | 0.0147 | Dominates |

ICER, incremental cost-effectiveness ratio; NHS, National Health Service; QALY, quality-adjusted life-year.

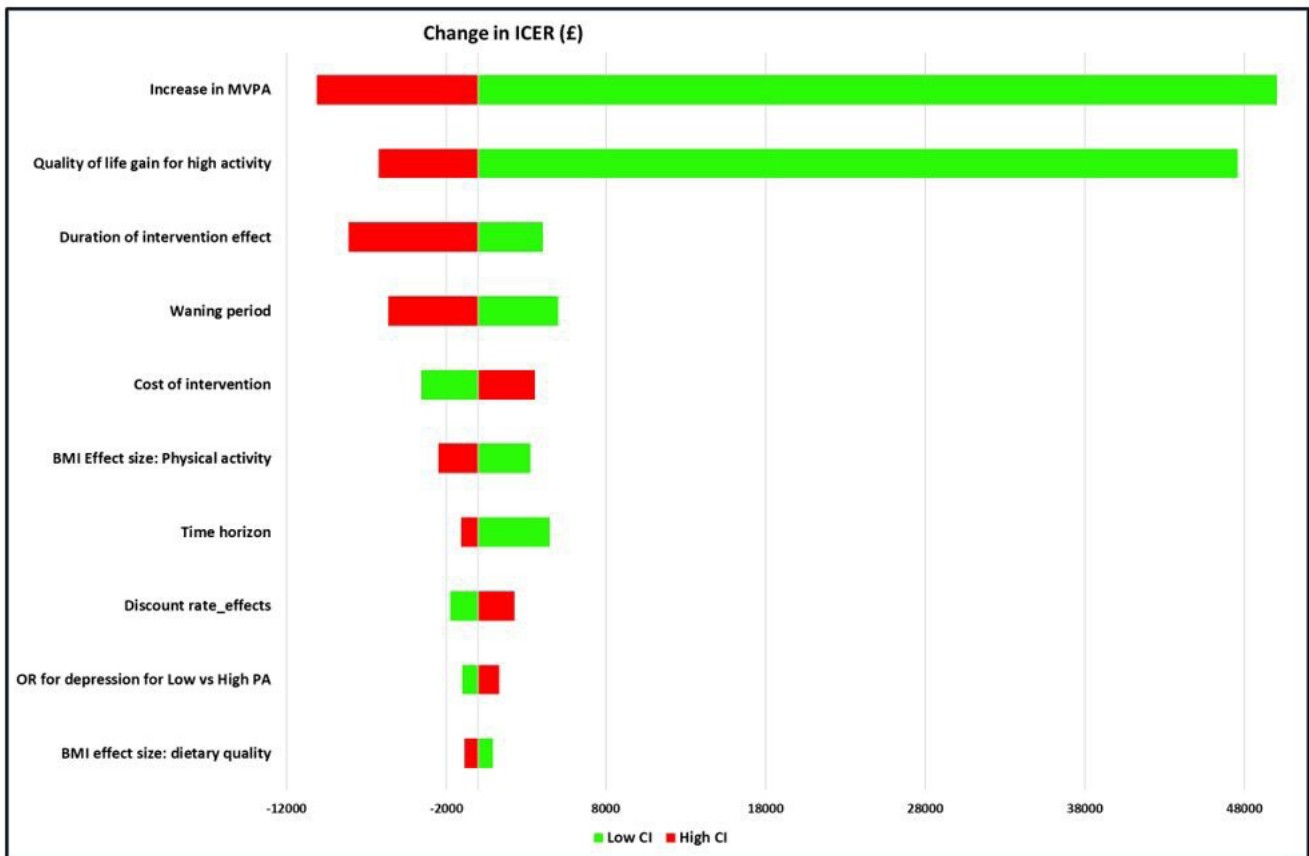

**Figure 2** Tornado diagram obtained from the deterministic sensitivity analysis showing the impact of varying the input values of the parameters that have the highest impact on the cost-effectiveness result. BMI, body massindex; ICER, Incremental cost effectiveness ratio; MVPA, moderate or vigorous physics activity.

would be less cost-effective (table 2). Changes to physical activity had a greater effect on the model results than changes in BMI. This is largely because physical activity is associated with an additional improvement in quality of life, independent of any health benefits associated with the health states in our model.

### Comparison with previous models

Previous cost-effectiveness studies evaluating dietary and physical activity interventions for adolescents have largely consisted of within-trial analyses that have not considered the benefits beyond the trial period.[57–61] Cost-effectiveness estimates vary between a cost saving of £408.22 (NZ$835) per child for a low intensity programme[57] to an additional cost of £120 630 per QALY gained for a more intensive intervention.[60] However, comparison between studies is difficult because of the differences in the study designs, the interventions considered and the outcomes reported.

Gc *et al*[1] developed a model to assess the long-term costs and health outcomes of two physical activity interventions targeting adolescents in the UK. The cost-effectiveness estimates varied from £11 426 per QALY for an after-school intervention to £68 056 per QALY for a multicomponent intervention. The costs of these interventions were estimated at £51 per participant for the after-school intervention and £207 ($A394) for the multicomponent

intervention. Their model included different health states to our model for diseases which typically affect people in later life including chronic heart disease, stroke, heart failure, breast failure and colorectal cancer. They ran the model for a lifetime horizon of 65 years. We have not included these health states as we adopted a conservative assumption that treatment benefits for adolescents from such multicomponent intervention as LifeLab Plus do not persist beyond 20 years.

### Strengths and limitations

The study addresses an important public health question by examining whether interventions targeting healthy eating and increased physical activity for adolescents provide value for money from a societal perspective. We incorporate existing evidence on the effect of improvement in adolescent health behaviours on four high prevalence short-to-medium term benefits relevant to adolescents: improved mental health, higher earnings, improved pregnancy outcomes and prevention oftype 2 diabetes. Sources for data used within our model were identified from a targeted literature review. Where data were not available for adolescents, we used data from adult population. Model structure and assumptions were informed by this review and discussions with public health experts. UK-specific incidence rates were used to ensure

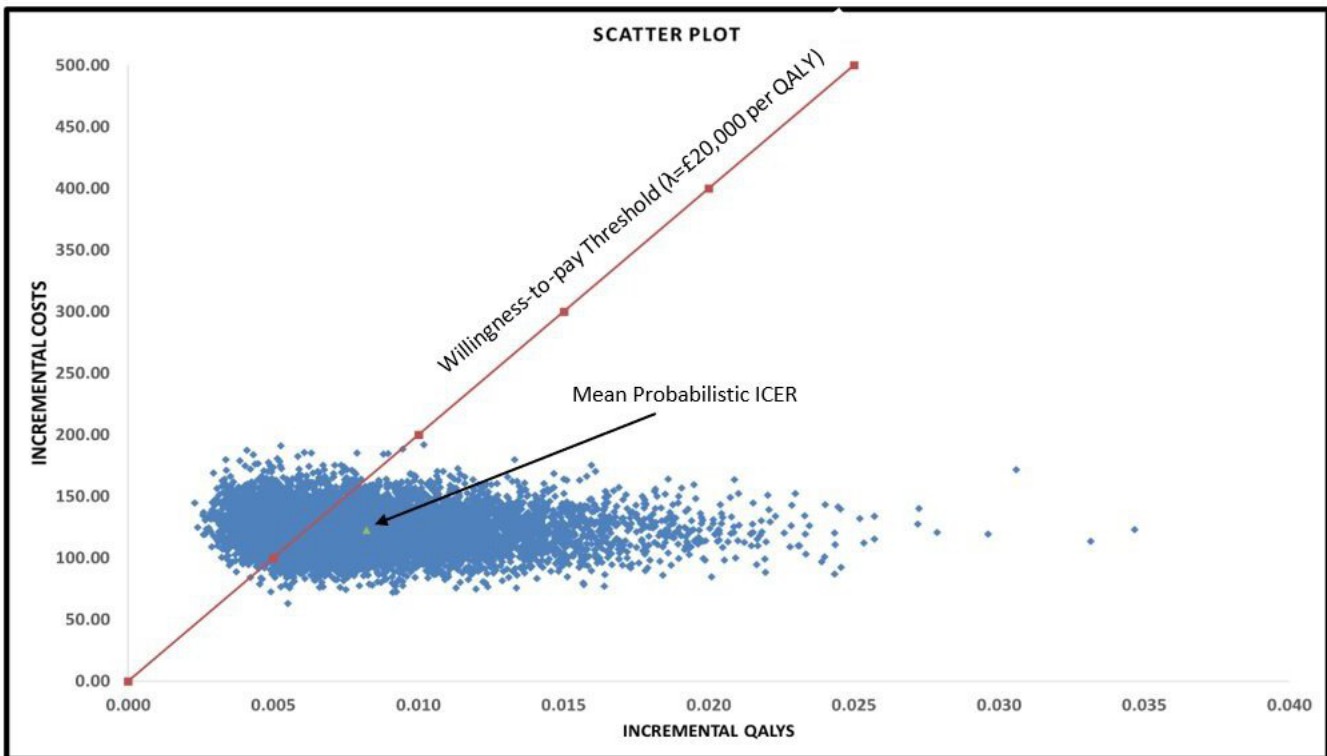

**Figure 3** Scatter plot obtained from the probabilistic sensitivity analysis. Results from all the 10 000 simulations fall within the North-East quadrant of the cost-effectiveness plane thereby indicating that the intervention (LifeLab Plus) is more expensive and more effective compared with usual schooling. Majority of the plots, including the mean ICER, fall within the willingness-to-pay threshold of £20 000 per QALY. PSA, probabilistic sensitivity analyses; QALYs, quality-adjusted life-year.

that patients entering the model matched the likely distribution of events in the UK. Our analysis focuses on a pragmatic subset of specific outcomes, while noting these could be expanded on with emerging data on effectiveness, notably in relation to mental health and other long-term conditions.

We do not attempt to estimate the long-term impacts of such interventions due to the lack of longitudinal data on lifetime trajectories of healthy diet and increased levels of physical activity. Our preliminary model suggests that this type of intervention is likely to be cost-effective based on short-to-medium term effects alone. If this is correct and the effects last longer, then additional benefits such as prevention of cardiovascular disease would enhance the cost-effectiveness of these interventions. This suggests that our cost-effectiveness results are conservative, and that quantification of long-term benefits is not necessary to demonstrate the cost-effectiveness of this type of intervention. However, the sensitivity analyses reported above do indicate some important uncertainties.

Our model has several shortcomings. The EACH-B study, which is the main trial evidence, is ongoing so there is currently no direct data on the effectiveness of this intervention. We have used evidence on treatment effectiveness from a systematic review of related interventions. When results become available from the EACH-B study, they will add to the evidence base and reduce uncertainty in these results. Further evidence for other model parameters

would also be valuable. For example, where data from adolescent age groups were not available, we used epidemiological estimates from adult populations that might not be transferable. There is uncertainty around key assumptions related to the duration of benefits from the intervention. In our analysis, we have assumed that the benefit observed in the clinical trials will last for 4 years and then will gradually reduce over the next 10 years. We only include costs and effects over a 20-year time horizon and have assumed that there would be no further benefits of the intervention for chronic diseases such as cardiovascular diseases and diabetes. There are also limitations in our approach to estimating income lost due to obesity. We have adopted a simple approach; however, this is a complex interacting bi-directional system. We have not fully explored whether income loss is due to obesity or whether obesity is caused by income loss through, for example, unemployment. Other broader uncertainties, such as long-term effects of COVID-19 on behaviour and mental health are also not addressed.

## Conclusion

Complex behavioural interventions that aim to improve diet and increase levels of physical activity among school-aged children have the potential to provide cost-effective use of UK healthcare resources. Such interventions have the potential to reduce burden of NCDs, although benefits in later life are more sensitive to assumptions about

the persistence of behavioural change and discounting. Our analysis focuses on a pragmatic subset of specific outcomes based on published literature on effectiveness of similar interventions, while noting these could be expanded on with emerging data on effectiveness, notably in relation to mental health and other long-term conditions.

**Author affiliations**
[1]Southampton Health Technology Assessments Centre, University of Southampton, Southampton, UK
[2]MRC Lifecourse Epidemiology Centre, University of Southampton, Southampton, UK
[3]NIHR Southampton Biomedical Research Centre, University Hospital Southampton NHS Foundation Trust, Southampton, UK
[4]Southampton Education School, Faculty of Social Sciences, University of Southampton, Southampton, UK
[5]Nuffield Department of Orthopaedics, Rheumatology and Musculoskeletal Sciences, University of Oxford, Oxford, UK
[6]School of Health Sciences, University of Southampton, Southampton, UK

**Collaborators** The LifeLab Team, University of Southampton, UK

**Contributors** All authors meet the criteria for authorship as set out in the submission guidance in the journal, and have read and approved the final version of the manuscript. NK designed the model, performed the analysis, interpreted the analysis, wrote the first draft of the manuscript and is responsible for the overall content as the guarantor. NK accepts full responsibility for the work and controlled the decision to publish. KC designed the model and performed the analysis. JL supervised the process, provided comments on model structure and data analysis. KW-T provided inputs for model parameters. JB, KG, HI, MB and CC provided critical comments on the model structure. All authors contributed to the critical revision of the manuscript and approved the final version of the manuscript.

**Funding** The research is funded by the National Institute for Health Research (RP-PG-0216-20004). HI and KG are funded by the UK Medical Research Council (MC_UU_12011/4). KG is supported by the National Institute for Health Research (NIHR Senior Investigator (NF-SI-0515-10042), NIHR Southampton 1000DaysPlus Global Nutrition Research Group (17/63/154) and NIHR Southampton Biomedical Research Centre (IS-BRC-1215-20004)), the European Union (Erasmus+ Programme ImpENSA 598488-EPP-1-2018-1-DE-EPPKA2-CBHE-JP) and the British Heart Foundation (RG/15/17/3174).

**Competing interests** KG has received reimbursement for speaking at conferences sponsored by companies selling nutritional products and is part of an academic consortium that has received research funding from Abbott Nutrition, Nestec, BenevolentAI Bio and Danone. The other authors have no potentially competing interests to declare.

**Patient and public involvement** Patients and/or the public were involved in the design, or conduct, or reporting, or dissemination plans of this research. Refer to the Methods section for further details.

**Patient consent for publication** Not applicable.

**Provenance and peer review** Not commissioned; externally peer reviewed.

**Data availability statement** All data relevant to the study are included in the article or uploaded as online supplemental information. Data from publicly available sources were used for the model. All model inputs are described in the paper and online supplemental material.

**ORCID iDs**
Neelam Kalita http://orcid.org/0000-0002-0973-0160
Keith Cooper http://orcid.org/0000-0002-0318-7670
Hazel Inskip http://orcid.org/0000-0001-8897-1749

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
