## [Reviewer comments · BMJ Open]

ARTICLE DETAILS

TITLE (PROVISIONAL)	Cost-effectiveness of a dietary and physical activity intervention in adolescents: a prototype modelling study based on the Engaging Adolescents in Changing Behaviour (EACH-B) programme
AUTHORS	Kalita, Neelam; Cooper, Keith; Baird, Janis; Woods-Townsend, Kathryn; Godfrey, Keith; Cooper, Cyrus; Inskip, Hazel; Barker, Mary; Lord, Joanne

VERSION 1 – REVIEW

REVIEWER	Guy Ludbrook Royal Adelaide Hospital
REVIEW RETURNED	28-Jun-2021

GENERAL COMMENTS	This manuscript presents the cost effectiveness analysis of interventions in adolescents to affect lifestyles choices and the subsequent impacts on certain outcomes. It applies Markov modelling and finds improved outcomes with additional cost, and cost cost-effectiveness below the usually accepted cost per QALY threshold. This field of healthcare is an important one, as diet and exercise are well-recognised as areas of concern, and the specific consequences studied are relevant and spread across specific medical conditions (diabetes, mental health, pregnancy outcomes) and measures of productivity (income earning). The introduction would be improved with the inclusion of specific aims and hypotheses of this paper – presumably they relate to a C-E model and its output. The focus on short- to medium-term outcomes is appropriate. In general, the thesis that interventions have an impact is supported by well referenced data, which is appropriately included and referenced. No attempt was made to examine longer term, but relevant, issues such as cardiovascular disease, which is logical considering the uncertainty about the sustainability of interventions on behaviours. How future data may allow this to be include in later modelling warrants discussion, especially as the long-term consequences on health, QALY and costs are substantial indeed. What specifically will arise from the EACH-B trial warrants clear description. The application of Markov modelling to areas such as this is important. Spend on healthcare interventions is going to need to be carefully managed in an era of increased demand and resource constraints. The model construction is well described, along with the pragmatic subset of specific outcomes. Again whether these can be expanded
---

	on with emerging data warrants discussion. For example, a broader spectrum of mental health disorders, or post-partum outcomes for children born considering the impact of eg diabetes on the newborn. These are within the time horizon chosen (20 years) and ultimately would broaden the cost-effectiveness calculations. The data to populate the model seem relevant, and are clearly outlined. In rapidly changing times, such as with Covid, a discussion on whether, and which of, these data may not fit future trends warrants discussion – mental health is an obvious example. I note that a hazard ratio for mortality of almost three was applied for diabetes in those <65 years. Is this relevant for a population with a mean age of 13 years and a model life cycle of 20 years, or does mortality risk increase later in life? Intervention cost data seem reasonably explained and justified, as does health state cost data. I note diabetes costs do not include complications as it is too early for these to develop. This is probably true, but does it reconcile well with the choice of a higher mortality chosen - risk ratio almost three? Are there any increased costs associated with children of parents with diabetes – see also the previous comment on outcomes of progeny? Cost-effectiveness analysis seems sound, with an appropriate threshold. For DSA sensitivity analysis, 95% CIs were used when available. What was used when they were not? For PSA, 1000 simulations were run. How was this number chosen? Would 10,000 be more informative? Scenario analysis to examine the sustainability of the intervention is logical. The distributions chosen (beta, gamma...) should be justified. The ICER findings support positive outcomes and increased cost – NE quadrant. Can the uncertainty of the ICER be provided? Would e.g. 10,000 simulations have changed the distribution outside the NE quadrant? The scenario analysis is well presented and clearly shows the impact of assumptions on ICER. The discussion cites a previous C-E model which included later health outcomes. The authors suggest they did not extend their model that far because of uncertainty on sustainability of outcomes. Is it plausible that even short term (<20 years) impacts on e.g. diabetes might influence later CVS complications? As mentioned previously, discussing what emerging data may improve the model would be valuable. For example, will EACH-B provide more certainty on the duration of effect of the intervention which clearly has a large impact on ICER? The conclusion is focussed on healthcare change as a result of an intervention. Inclusion of specific study aims (see before) which presumably are around successful development of a model and initial ICER estimations, would better fit with a conclusion based around C-E modelling.
--	--

REVIEWER	Fei Men The University of Alabama, Consumer Sciences
REVIEW RETURNED	23-Nov-2021

GENERAL COMMENTS	This is a well written paper addressing a policy relevant question. My comments are listed below. - What is the theoretical basis for the base case assumption that the treatment effect would sustain for 4 years and wane over a 10-year period? According to Table 2, it seems that a slight change in
---

	assumption could move the cost past the 20,000-pound threshold. - Figure 2 showed that dietary quality and physical activity had negligible impact on ICER through the mediation of BMI. Since such mediating effect was one of the key hypotheses of the study, I'd appreciate more discussion on this finding and the authors' interpretation of Figure 2. - I am struggling to see a clear trend or pattern in the scatter plot (Figure 3). The authors could help readers understand the implication of the figure by adding trend line and slope values.
--	--

REVIEWER	Flavia Sarti University of Sao Paulo, Nutrition in Public Health
REVIEW RETURNED	17-Dec-2021

GENERAL COMMENTS	The manuscript presents interesting approach to cost-effectiveness analysis of intervention for health promotion among adolescents in the school environment, i.e., economic assessment of the Engaging Adolescents in Changing Behaviour (EACH-B) programme through Markov model. However, the general quality of the text is inadequate for publication (e.g., several short sentences like line 105, page 5; line 111, page 4, etc.), and there are numerous errors in references throughout the manuscript (e.g., line 175, page 6). In addition, the following problems should be addressed by the authors to ensure proper presentation of the context and adequate description of the study: (1) The abstract should be revised to include: 1a. Further details on the simulations performed (including size of the simulated cohort, line 38, page 2); 1b. Appropriate indication of the health outcomes that were targeted in the analysis (instead of generic mentions to "better mental health, Type 2 diabetes, higher earnings and reduced incidences of adverse pregnancy outcomes", lines 33-34, page 2); in fact, the actual measures that comprise the health outcomes of the model are described only in line 245, page 9; 1c. Presentation of the results of the study (instead of general remarks on the acceptability of the intervention at a willingness-to-pay threshold of 20,000 pounds per QALY, lines 45-46, page 2); (2) The introduction should be revised to: 2a. Start with the general context of social, health and economic impacts of non-communicable diseases in the UK or worldwide (page 4, line 96); 2b. Indicate the location of the estimates presented in page 5, lines 121-122 (UK? Worldwide?); 2c. Reconsider the sentence on the literature gap mentioned in page 5, lines 141-143 ("there is sparse economic evidence assessing health effects and costs of such interventions"), considering the abundance of studies in the subject (e.g., https://doi.org/10.1038/sj.ijo.0803469 ; https://doi.org/10.1371/journal.pmed.1003210 ; https://doi.org/10.1016/j.jadohealth.2017.05.024 ; https://doi.org/10.1016/j.nut.2011.11.016 ; https://doi.org/10.1111/j.1746-1561.2008.00357.x ; https://doi.org/10.1016/j.acap.2015.07.009 ; and the study cited in the text: http://dx.doi.org/10.1136/bmjopen-2017-018640); further contradicted by the remarks mentioned in lines 413-414, page 13 ("Previous cost-effectiveness studies evaluating dietary and physical activity interventions for adolescents");
---

	(3) The section of methods requires in-depth reformulation: 3a. Include the actual health outcomes targeted in the Markov model in page 6, lines 163-164 (e.g., which measures of mental health? which adverse pregnancy outcomes?); 3b. Include the cohort size (page 6, line 178) in the Markov model; 3c. Describe adequately the simulation mechanisms, including proper description of the health states (No Type 2 Diabetes, Type 2 Diabetes and Death), transitions between health states, and relations with other health outcomes (mental health, loss of earnings and adverse pregnancy outcome), since it is unclear whether the other health outcomes are directly related to occurrence of diabetes or independent of the occurrence of diabetes (lines 184-187, page 6; lines 198-199, page 7; line 202, page 7); 3d. Improve the description of the intervention, mechanisms of persistence of effects and time horizon (lines 209-212, page 7): the cohort is variable (i.e., certain number of adolescents enter 13 years-old and engage in the intervention, then complete 14 years-old and are excluded from the intervention, and another cohort of 13-years-old adolescents enter the intervention?) or fixed (there is only one cohort of 13-years-old adolescents that go through the intervention and after the intervention there are waning effects of the intervention?); the duration of the intervention in the Markov model is one year to include only 13 years-old adolescents?; 3e. Revise the actual recommendation of physical activity level for children and adolescents, lines 260-261, page 9 (300 minutes per week, according to https://www.nhs.uk/live-well/exercise/physical-activity-guidelines-children-and-young-people/ and https://ijbnpa.biomedcentral.com/articles/10.1186/s12966-020-01037-z); 3f. Provide detailed description of direct and indirect costs included in the model, line 315, page 10 (preferably including a table synthesizing resources, costs, and sources of information; instead of the long and confusing description in the text); (4) The discussion of the manuscript should be revised to: 4a. Provide proper comparison between cost-effectiveness ratios of interventions identified in other studies by converting different currencies into international currency using purchase power parity conversion factors in a single reference period (lines 415-471, page 13; lines 424-425, page 14); 4b. Indicate that long-term models are sensitive to changes in risk to other chronic diseases (like cardiovascular risk) due to differences in assumptions regarding the persistence of behavioral change, and also due to potential adverse health outcomes from intense physical activity (e.g., contusions); (5) Figure 1 should be revised to exclude transitions from health states referring to "death", since it makes no sense in "maintaining" death in sequential periods (i.e., once an individual dies, he/she should be excluded from the model, thus, there is no point in including circular arrows from a state of "death" into itself).
--	---

VERSION 1 – AUTHOR RESPONSE

Comments	Authors' response
The manuscript presents interesting approach to cost-effectiveness analysis of intervention for health promotion among adolescents in the school environment, i.e., economic assessment of the Engaging Adolescents in Changing Behaviour (EACH-B) programme through Markov model. However, the general quality of the text is inadequate for publication (e.g., several short sentences like line 105, page 5; line 111, page 4, etc.), and there are numerous errors in references throughout the manuscript (e.g., line 175, page 6).	Thank you for the feedback. We re-checked the references and proof-read the manuscript, following which we have revised texts throughout the manuscript (all saved as track changes) to improve clarity and readability.
1. The abstract should be revised to include:	Please see below the responses to individual comments.
1a. Further details on the simulations performed (including size of the simulated cohort, line 38, page 2);	We have revised the abstract to include the number of simulations (within 'Design') and the cohort size (within 'Participants').
1b. Appropriate indication of the health outcomes that were targeted in the analysis (instead of generic mentions to "better mental health, Type 2 diabetes, higher earnings and reduced incidences of adverse pregnancy outcomes", lines 33-34, page 2); in fact, the actual measures that comprise the health outcomes of the model are described only in line 245, page 9;	Revised the abstract to reflect the reviewer's comment within 'Design'.
1c. Presentation of the results of the study (instead of general remarks on the acceptability of the intervention at a willingness-to-pay threshold of 20,000 pounds per QALY, lines 45-46, page 2);	Revised the abstract to include the cost effectiveness results within 'Results'.
2. The introduction should be revised to:	
2a. Start with the general context of social, health and economic impacts of non-communicable diseases in the UK or worldwide (page 4, line 96);	Thank you for the feedback. Whilst we have noted this, we haven't revised the text as we view this as a stylistic preference.
2b. Indicate the location of the estimates presented in page 5, lines 121-122 (UK? Worldwide?);	We have revised the text to include 'in the UK' (line 120)
2c. Reconsider the sentence on the literature gap mentioned in page 5, lines 141-143 ("there is sparse economic evidence assessing health effects and costs of such interventions"), considering the abundance of studies in the	We have revised the text to reflect the reviewer's comment and cited sources in lines 139 and 140.

subject (e.g., https://doi.org/10.1038/sj.ijo.0803469 ; https://doi.org/10.1371/journal.pmed.1003210 ; https://doi.org/10.1016/j.jadohealth.2017.05.024 ; https://doi.org/10.1016/j.nut.2011.11.016 ; https://doi.org/10.1111/j.1746-1561.2008.00357.x ; https://doi.org/10.1016/j.acap.2015.07.009 ; and the study cited in the text: http://dx.doi.org/10.1136/bmjopen-2017-018640); further contradicted by the remarks mentioned in lines 413-414, page 13 ("Previous cost-effectiveness studies evaluating dietary and physical activity interventions for adolescents");	
The section of methods requires in-depth reformulation: 3a. Include the actual health outcomes targeted in the Markov model in page 6, lines 163-164 (e.g., which measures of mental health? which adverse pregnancy outcomes?);	We have revised the text to include the following statement in lines 155 to 159: 'The model focussed on four potential short to medium-term benefits of healthy eating and physical activity in this age group: better mental health (episodes of depression and generalised anxiety disorder), outcomes, higher earnings and reduced incidences of Type 2 diabetes and adverse pregnancy outcomes (in terms of pre-term delivery)'
3b. Include the cohort size (page 6, line 178) in the Markov model	Cohort size has been included in line 173.
3c. Describe adequately the simulation mechanisms, including proper description of the health states (No Type 2 Diabetes, Type 2 Diabetes and Death), transitions between health states, and relations with other health outcomes (mental health, loss of earnings and adverse pregnancy outcome), since it is unclear whether the other health outcomes are directly related to occurrence of diabetes or independent of the occurrence of diabetes (lines 184-187, page 6; lines 198-199, page 7; line 202, page 7);	We have revised the texts within the subsection 'Model states' in the Methods section. The changes are lines 178 to 190.
3d. Improve the description of the intervention, mechanisms of persistence of effects and time horizon (lines 209-212, page 7): the cohort is variable (i.e., certain number of adolescents enter 13 years-old and engage in the intervention, then complete 14 years-old and are excluded from the intervention, and another cohort of 13-years-old adolescents enter the intervention?) or fixed (there is only one cohort of 13-years-old adolescents that go through the intervention and after the intervention there are waning effects of the intervention?); the duration of the intervention in the Markov model is one year to include only 13 years-old adolescents?;	We have revised the description within the subsection 'persistence of effects and time horizon' in the Methods section, in lines 210 to 220.

3e. Revise the actual recommendation of physical activity level for children and adolescents, lines 260-261, page 9 (300 minutes per week, according to https://eur03.safelinks.protection.outlook.com/?url=https%3A%2F%2Fwww.nhs.uk%2Flive-well%2Fexercise%2Fphysical-activity-guidelines-children-and-young-people%2F&data=04%7C01%7CN.Kalita%40soton.ac.uk%7Cf4a7ec3ee9074d36352f08d9c15402f3%7C4a5378f929f44d3ebe89669d03ada9d8%7C0%7C0%7C637753388834943537%7CUnknown%7CTWFpbGZsb3d8eyJWljiMC4wLjAwMDAilCJQljoiv2luMzliLjBTiil6lk1haWwiLCJXVCI6Mn0%3D%7C3000&data=HtlFrwclVMmUpzNbPKEcCRgghUhXNOz82BvGki3KAoY%3D&reserved=0 and https://eur03.safelinks.protection.outlook.com/?url=https%3A%2F%2Fijbnpa.biomedcentral.com%2Farticles%2F10.1186%2Fs12966-020-01037-z&data=04%7C01%7CN.Kalita%40soton.ac.uk%7Cf4a7ec3ee9074d36352f08d9c15402f3%7C4a5378f929f44d3ebe89669d03ada9d8%7C0%7C0%7C637753388834943537%7CUnknown%7CTWFpbGZsb3d8eyJWljiMC4wLjAwMDAilCJQljoiv2luMzliLjBTiil6lk1haWwiLCJXVCI6Mn0%3D%7C3000&data=6hc0ZA0Kn3OL5IKOJ1K%2FdYij2AOquwrGw0R4ePQuk7Y%3D&reserved=0);	We have revised the text as follows in lines 268 to 271: ‘Furthermore, an increase from being inactive to achieving 150 minutes of moderate-intensity physical activity per week was assumed to lower the risk of Type 2 diabetes incidence by 26%, after adjustment for body weight.’
3f. Provide detailed description of direct and indirect costs included in the model, line 315, page 10 (preferably including a table synthesizing resources, costs, and sources of information; instead of the long and confusing description in the text);	Table 1 in the manuscript summarises the costs included in the model. For further clarity, we have revised this table to include the sources. We view that the texts provide further description of the assumptions and methods used to arrive at the costs.
The discussion of the manuscript should be revised to: 4a. Provide proper comparison between cost-effectiveness ratios of interventions identified in other studies by converting different currencies into international currency using purchase power parity conversion factors in a single reference period (lines 415-471, page 13; lines 424-425, page 14);	As the price years (2014-16) are similar between studies and the countries have similar purchasing power, we have converted the costs into UK £ using the current exchange rate. £1 = 2.045 NZ\$, £1 = 1.904 AUS\$.
4b. Indicate that long-term models are sensitive to changes in risk to other chronic diseases (like cardiovascular risk) due to differences in assumptions regarding the persistence of behavioral change, and also due to potential adverse health outcomes from intense physical activity (e.g., contusions).	We agree that long-term models are sensitive to changes in risk to other chronic diseases arising from differences in assumptions regarding the persistence of behavioural change, alongside potential effects of moderating factors (such as later physical activity). However, for the purpose of this manuscript we focus on short to medium term impacts for reasons cited within the subsection ‘Persistence of effects and time horizon’. As stated earlier in this document, we

	view that the long-term extrapolation is not necessary if a simpler short-medium term extrapolation can show that the intervention is likely to be cost-effective. We clarify this stance in Discussion section in lines 447 to 455.
(5) Figure 1 should be revised to exclude transitions from health states referring to "death", since it makes no sense in "maintaining" death in sequential periods (i.e., once an individual dies, he/she should be excluded from the model, thus, there is no point in including circular arrows from a state of "death" into itself).	We have revised Figure 1 as per the reviewer's suggestion.

VERSION 2 – REVIEW

REVIEWER	Guy Ludbrook Royal Adelaide Hospital
REVIEW RETURNED	14-Mar-2022

GENERAL COMMENTS	This paper develops a Markov cost-effectiveness (C-E) model to estimate whether interventions such as that provided by the EACH-B trial may fall under the cost threshold for QALY improvement. The use of this modelling to evaluate such programmes is a positive one, although the fact this is a 'complex interacting bi-directional system' does raise the issue of uncertainty when using published data and expert opinion, rather than the actual trial data. The revised manuscript has been improved through the responses to reviewers' comments. A few specific comments are made below: The title needs to clearly define that this study uses only cost data from the EACH-B programme and effectiveness estimates for similar interventions from published literature. The authors use the term 'prototype model' in the text, which might be an appropriate title inclusion as, in some ways, this model is an hypothesis for a model, which will be evaluable once trial data are available. As stated in the text, the EACH-B results may provide better parameter estimates for the model. As an indication, what would be the title of the mentioned follow up modelling paper with EACH-B results? The pattern of treatment duration of effect on behaviour is identifiable as an experts' assumption, in contrast to many other parameters, and may be quite uncertain. This can be in part addressed once the EACH-B trial is complete. It is helpful that alternative assumptions about the persistence of effects on behaviour after trial follow-up have been explored. The probability of LifeLab Plus being cost-effective at the defined WTP was 69% compared with usual schooling. This is a relatively modest figure in scientific terms - it would be interesting to speculate the degree of programme impact that would be needed to provide a high level of certainty, and how this relates to the actual study's power calculations.
---

REVIEWER	Flavia Sarti University of Sao Paulo, Nutrition in Public Health
REVIEW RETURNED	15-Mar-2022

GENERAL COMMENTS	The manuscript proposes to perform a cost-effectiveness analysis of intervention for health promotion among adolescents in the school environment, i.e., economic assessment of the Engaging Adolescents in Changing Behaviour (EACH-B) programme through Markov model. The paper was substantially improved in relation to its initial version, and the results are interesting in terms of public policies of health. However, there are still minor details missing in the manuscript, described below: (1) Authors mention in the response to reviewers that "for the purpose of this manuscript we focus on short to medium term impacts"; however, impacts of the intervention on individuals' earnings seemed to comprise a long-term outcome (Table 1, Parameter equations: loss of earnings are described as units of £ over lifetime). In addition, there are no explanations on the assumptions referring to BMI influencing individuals' earnings in the case of changes in BMI throughout time, considering that the model assumes loss of earnings as transient state; (2) I did not find mentions to the software or computation resources (e.g., R, Python, MatLab or others) used to implement the simulations of the Markov model.
---

VERSION 2 – AUTHOR RESPONSE

Reviewer 1: Prof Guy Ludbrook

Comments	Authors' response
1. The title needs to clearly define that this study uses only cost data from the EACH-B programme and effectiveness estimates for similar interventions from published literature. The authors use the term 'protype model' in the text, which might be an appropriate title inclusion as, in some ways, this model is an hypothesis for a model, which will be evaluable once trial data are available. As stated in the text, the EACH-B results may provide better parameter estimates for the model. As an indication, what would be the title of the mentioned follow up modelling paper with EACH-B results?	The title has been revised to 'Cost-effectiveness of a dietary and physical activity intervention in adolescents: a prototype modelling study based on the Engaging Adolescents in Changing Behaviour (EACH-B) programme'
2. The pattern of treatment duration of effect on behaviour is identifiable as an experts' assumption, in contrast to many other parameters, and may be quite uncertain. This can be in part addressed once the EACH-B trial is complete. It is helpful that alternative assumptions about the persistence of effects on behaviour after trial follow-up have been explored.	Thank you, we have noted the feedback. No change to text.
3. The probability of LifeLab Plus being cost-effective at the defined WTP was 69% compared with usual schooling. This is a relatively modest figure in scientific terms - it would be interesting to speculate the degree of programme impact that would be needed to provide a high level of certainty, and how this relates to the actual study's power calculations.	It is not generally thought appropriate to apply a fixed level of statistical significance (e.g. 95%) to cost-effectiveness results. See Karl Claxton's 'Irrelevance of inference' paper for an explanation of the rationale for this (DOI: 10.1016/s0167-296(98)00039-3). Instead, an estimation approach is taken, and probabilities of cost-effectiveness are presented so that

	decision makers can take account of uncertainty alongside other considerations when weighing up funding decision (e.g. see section 4.7 in the NICE health technology evaluation manual). We will revise the estimated probability of cost-effectiveness for the EACH-B intervention when the final trial results are available. As for most clinical trials, the sample size calculations for EACH-B were based on the primary clinical outcome, not cost-effectiveness.
--	---

Reviewer 3: Prof Flavia Sarti

Comments	Authors' response
1. Authors mention in the response to reviewers that "for the purpose of this manuscript we focus on short to medium term impacts"; however, impacts of the intervention on individuals' earnings seemed to comprise a long-term outcome (Table 1, Parameter equations: loss of earnings are described as units of £ over lifetime). In addition, there are no explanations on the assumptions referring to BMI influencing individuals' earnings in the case of changes in BMI throughout time, considering that the model assumes loss of earnings as transient state.	The loss of earnings, which is applied to change in BMI over time is estimated annually over the model time horizon. It is scaled according to the population age to factor in average income by age. We have inserted the above texts within lines 262 to 264 in the manuscript.
2. I did not find mentions to the software or computation resources (e.g., R, Python, MatLab or others) used to implement the simulations of the Markov model.	The Markov model was developed and implemented in Microsoft Excel, including all the model simulations. This is stated in line 371 of the manuscript.

VERSION 3 – REVIEW

REVIEWER	Guy Ludbrook Royal Adelaide Hospital
REVIEW RETURNED	03-Jun-2022

GENERAL COMMENTS	This manuscript outlines well a prototype model to determine cost-effectiveness of a behaviour intervention. The addition of prototype to the title, and the qualifications on the results in the text appear to adequately outline the opportunities, and limitations, of the this model and the data available to date. I would suggest a small change. The Abstract Conclusion should be adjusted to better align with that in the manuscript text - as stands it isn't really a conclusion but an explanation of some of the findings. For example, the sentence "The results suggested that such an intervention has the potential to offer a cost-effective use of healthcare-resources for adolescents in the UK at a willingness-to-
---

	pay threshold of £20,000 per QALY" in the Abstract Results appears more suitable to a Conclusion.
--	---

REVIEWER	Flavia Sarti University of Sao Paulo, Nutrition in Public Health
REVIEW RETURNED	25-May-2022

GENERAL COMMENTS	The manuscript focuses on an interesting subject in the field of public health, i.e., assessment of costs, health outcomes and cost-effectiveness of interventions aimed at improving the diet quality and the physical activity level of adolescents through computational modelling. The revised version of the manuscript clarified the issues I presented in my last review, therefore, the manuscript is acceptable for publication in its current version.
--

VERSION 3 – AUTHOR RESPONSE

Reviewer 1: Prof Guy Ludbrook

Comments	Authors' response
This manuscript outlines well a prototype model to determine cost-effectiveness of a behaviour intervention. The addition of prototype to the title, and the qualifications on the results in the text appear to adequately outline the opportunities, and limitations, of this model and the data available to date. I would suggest a small change. The Abstract Conclusion should be adjusted to better align with that in the manuscript text - as stands it isn't really a conclusion but an explanation of some of the findings. For example, the sentence "The results suggested that such an intervention has the potential to offer a cost-effective use of healthcare-resources for adolescents in the UK at a willingness-to-pay threshold of £20,000 per QALY" in the Abstract Results appears more suitable to a Conclusion.	The Abstract results and conclusion have been revised as follows. “Results The exemplar dietary and physical activity intervention was associated with an incremental cost of £123 per adolescent and better health outcomes with a mean QALY gain of 0.0085 compared with usual schooling, resulting in an ICER of £14,367 per QALY. The key model drivers are the intervention effect on levels of physical activity, quality-of-life gain for high levels of physical activity, the duration of the intervention effects and the period over which effects wane. Conclusions The results suggested that such an intervention has the potential to offer a cost-effective use of healthcare-resources for adolescents in the UK at a willingness-to-pay threshold of £20,000 per QALY. The model focused on short to medium-term benefits of healthy eating and physical activity exploiting the strong evidence base that exists for this age group. Other benefits in later life, such as reduced cardiovascular risk, are more sensitive to assumptions about the persistence of behavioural change and discounting.”

Reviewer 3: Prof Flavia Sarti

Comments	Authors' response
The manuscript focuses on an interesting subject in the field of public health, i.e., assessment of costs, health	Thank you. We have noted the feedback.

outcomes and cost-effectiveness of interventions aimed at improving the diet quality and the physical activity level of adolescents through computational modelling. The revised version of the manuscript clarified the issues I presented in my last review, therefore, the manuscript is acceptable for publication in its current version.	
---	--